# An MHD Fluid Flow over a Porous Stretching/Shrinking Sheet with Slips and Mass Transpiration

**DOI:** 10.3390/mi13010116

**Published:** 2022-01-12

**Authors:** A. B. Vishalakshi, U. S. Mahabaleshwar, Ioannis E. Sarris

**Affiliations:** 1Department of Mathematics, Davangere University, Shivagangothri, Davangere 577 007, India; vishalavishu691@gmail.com; 2Department of Mechanical Engineering, University of West Attica, 250 Thivon and P. Ralli Str., 12244 Athens, Greece; ioannisarris@gmail.com

**Keywords:** porous sheet, MHD, three-dimensional, thermal efficiency, exact solution, nanofluid

## Abstract

In the present paper, an MHD three-dimensional non-Newtonian fluid flow over a porous stretching/shrinking sheet in the presence of mass transpiration and thermal radiation is examined. This problem mainly focusses on an analytical solution; graphene water is immersed in the flow of a fluid to enhance the thermal efficiency. The given non-linear PDEs are mapped into ODEs via suitable transformations, then the solution is obtained in terms of incomplete gamma function. The momentum equation is analyzed, and to derive the mass transpiration analytically, this mass transpiration is used in the heat transfer analysis and to find the analytical results with a Biot number. Physical significance parameters, including volume fraction, skin friction, mass transpiration, and thermal radiation, can be analyzed with the help of graphical representations. We indicate the unique solution at stretching sheet and multiple solution at shrinking sheet. The physical scenario can be understood with the help of different physical parameters, namely a Biot number, magnetic parameter, inverse Darcy number, Prandtl number, and thermal radiation; these physical parameters control the analytical results. Graphene nanoparticles are used to analyze the present study, and the value of the Prandtl number is fixed to 6.2. The graphical representations help to discuss the results of the present work. This problem is used in many industrial applications such as Polymer extrusion, paper production, metal cooling, glass blowing, etc. At the end of this work, we found that the velocity and temperature profile increases with the increasing values of the viscoelastic parameter and solid volume fraction; additionally, efficiency is increased for higher values of thermal radiation.

## 1. Introduction

Transport is one of the main motivations for conducting an experiment on stretching sheet problems, the main reason this problem is widely utilized in the engineering and industrial processes viz., extrusion of sheet, metal thinning, exchange of heat between, etc. Sakiadis [1,2] was the first researcher to investigate the stretching sheet problem. Later, Crane [3] elaborated this problem with flow past a stretching sheet. Motivated by Crane’s work, many researchers conducted experiments on stretching sheet problems by observing their huge applications. Turkyilmazoglu [4] investigated the impact of MHD on thermal slip flow due to stretching sheet. Mahabaleshwar et al. [5,6,7] conducted experiments with magneto hydrodynamics in the presence of different physical parameters such as mass transpiration, thermal radiation, and Dufour and Soret mechanisms. Later, these investigations were carried out with porous sheet; these porous particles play a major role in stretching sheet problems as these porous materials help to enrich the rate of heat transfer from stretching/shrinking surfaces and have numerous industrial applications. Therefore, many researchers show interest in investigating porous media. Elboshbeshy and Bazid [8,9] investigated variable viscosity fluid flow with a porous medium and internal heat generation. Further, this work was extended by Cortell [10]; he included power law temperature distribution. Mahabaleshwar et al. [11,12,13,14] demonstrated many porous stretching sheet problems with different flow fluids and a variety of boundary conditions. Rasool et al. [15] worked on second grade nanofluid flow with the Darcy–Forchheimer medium in the presence of thermal radiation and viscous dissipation (see further recent works on nanofluids in [16,17,18]).

Apart from these discoveries, some experiments take place with nanofluids, as these nanofluids offer a better thermal efficiency than base fluids. Rahman et al. [19] examined the nanofluid flow with porous exponential on the basis of Buongiorno’s method. Mahabaleshwar et al. [20] and Benos et al. [21] examined the problem statement analytically in the presence of nanofluid. Shafiq et al. [22] investigated thermally enhanced Darcy–Forchheimer Casson-water/Glycerin rotating nanofluids by using the effect of uniform magnetic field. Rasool et al. [23] worked on MHD nanofluid flow on the basis of numerical scrutinization of Darcy–Forchheimer relation bound by nonlinear stretching surface with heat and mass transfer. Rasool and Shafiq [24] explained the Darcy medium with thermally enhanced chemically reactive Powell–Eyring nanofluid flow over a non-linearly stretching surface affected by a transverse magnetic field with convective boundary conditions. Afridi et al. [25] studied the 3-D dissipative flow with hybrid nanofluid; in this work, thermophysical models were used to investigate the entropy generation. Dianchen Lu et al. [26] examined the entropy generation by considering dissipative nanofluid flow by using the effect of magnetic dissipation and transpiration. Graphene is one of the most useful nanomaterials as it has an extraordinary blend of superb properties, better heat and electrical conduction, and optical transparency compared to other materials. Additionally, graphene is the thinnest as well as the strongest material. Some recent works on graphene are given in [27,28,29].

The present work is motivated by the work of Turkyilmazoglu [30], explaining the analytical solution of 3-D flow of a fluid with magneto hydrodynamics by using various physical parameters. Motivated by the abovementioned articles, the present work explains the three-dimensional flow of a non-Newtonian fluid due to porous stretching/shrinking sheet. Magneto hydrodynamics and graphene water nanoparticles are also immersed in the flow of fluid to achieve better thermal efficiency. The given PDEs are converted into ODEs by using similarity variables. Incomplete gamma function is obtained at the end of the solution. By using various physical parameters, the problem is verified exactly, and the skin friction coefficient is examined. The novelty of the present work is to examine the problem analytically and to find the domain in terms of mass transpiration; this is used in the heat transfer to analyze the heat equation. The work is used in many industrial and engineering applications, viz., power engines, advanced nuclear systems, automobiles, biological sensors, drug delivery, and entropy generation [31].

### Problem Statement

An MHD graphene water nanofluid flow through a porous medium with mass transpiration and thermal radiation is considered. The physical diagram of the given problem is represented in Figure 1. Three-dimensional flow is subjected to wall temperature $T_{w}$ in a porous medium and a far-field temperature $T_{\infty }$. The porous medium is filled with graphene water nanoparticles. The quantities of nanofluid are indicated in Table 1. The equations of the fluid flow can be written in the form (see Mahabaleshwar et al. [32], Siddheshwar et al. [33] and Riaz et al. [34]).
(1)$$\frac{\partial u}{\partial x}+\frac{\partial v}{\partial y}+\frac{\partial w}{\partial z}=0$$
(2)$$\begin{array}{ll} u\frac{\partial u}{\partial x}+v\frac{\partial u}{\partial y} & +w\frac{\partial u}{\partial z}=\nu_{nf}\frac{\partial^{2}u}{\partial z^{2}}+\gamma_{0}u-\left(\frac{\mu_{nf}}{K\rho_{nf}}+\frac{\sigma_{nf}B_{0}{}^{2}}{\rho_{nf}}\right)u\\ & k\left\{u\frac{\partial^{3}u}{\partial x\partial z^{2}}+w\frac{\partial^{3}u}{\partial z^{3}}-\left(\frac{\partial u}{\partial x}\frac{\partial^{2}u}{\partial z^{2}}+\frac{\partial u}{\partial z}\frac{\partial^{2}w}{\partial z^{2}}+2\frac{\partial u}{\partial z}\frac{\partial^{2}u}{\partial x\partial z}+2\frac{\partial w}{\partial z}\frac{\partial^{2}u}{\partial z^{2}}\right)\right\} \end{array}$$
(3)$$\begin{array}{ll} u\frac{\partial v}{\partial x}+v\frac{\partial v}{\partial y} & +w\frac{\partial v}{\partial z}=\nu_{nf}\frac{\partial^{2}v}{\partial z^{2}}+\gamma_{0}v-\left(\frac{\mu_{nf}}{K\rho_{nf}}+\frac{\sigma_{nf}B_{0}{}^{2}}{\rho_{nf}}\right)v\\ & k\left\{v\frac{\partial^{3}v}{\partial y\partial z^{2}}+w\frac{\partial^{3}v}{\partial z^{3}}-\left(\frac{\partial v}{\partial y}\frac{\partial^{2}v}{\partial z^{2}}+\frac{\partial v}{\partial z}\frac{\partial^{2}w}{\partial z^{2}}+2\frac{\partial v}{\partial z}\frac{\partial^{2}v}{\partial y\partial z}+2\frac{\partial w}{\partial z}\frac{\partial^{2}v}{\partial z^{2}}\right)\right\} \end{array}$$
(4)$$u\frac{\partial T}{\partial x}+v\frac{\partial T}{\partial y}+w\frac{\partial T}{\partial z}=\frac{\kappa_{nf}}{{\left(\rho C_{P}\right)}_{nf}}\frac{\partial^{2}T}{\partial y^{2}}-\frac{1}{{\left(\rho C_{P}\right)}_{nf}}\frac{\partial q_{r}}{\partial y}$$
Subjected to appropriate boundary conditions are:(5)$$\left.\begin{array}{l} u=dax+l\frac{\partial u}{\partial z},\;\;v=by+l\frac{\partial v}{\partial z},\;\;w=w_{0},\;\;\;-\kappa \frac{\partial T}{\partial z}=h\left(T_{w}-T\right),\;\;\;\mathrm{at}\;\;\;\;z=0\\ u\to 0,\;\;\;\frac{\partial u}{\partial z}\to 0,\;\;\;v\to 0,\;\;\;\frac{\partial v}{\partial z}\to 0,\;\;\;\;T\to T_{\infty }\;\;\;\;\;\;\;\;\;\;\;\;\;\;\;\;\;\;\;\;as\;\;\;z\to \infty \end{array}\right\}$$

The heat flux $q_{r}$ can be defined on the basis of Rosseland’s approximation as follows (see refs. [35,36,37,38,39]):(6)$$q_{r}=-\frac{4\sigma^{*}}{3k^{*}}\frac{\partial T^{4}}{\partial y}\;$$
here, $\sigma^{*}$ is the Stefan–Boltzmann constant, $k^{*}$ is the coefficient of mean absorption, and *T* is the temperature of the fluid. 

Ambient temperature *T*^4^ expands in terms of Taylor’s series as
(7)$$T^{4}=T_{\infty }{}^{4}+4T_{\infty }{}^{3}\left(T-T_{\infty }\right)+6T_{\infty }{}^{2}{\left(T-T_{\infty }\right)}^{2}+....$$

In Equation (7), we ignore higher order terms to yield the equation as
(8)$$T^{4}=-3T_{\infty }{}^{4}+4T_{\infty }{}^{3}T$$

On applying Equation (8) into Equation (6), the first order derivative of heat flux can be given by
(9)$$\frac{\partial q_{r}}{\partial y}=-\frac{16\sigma^{*}T_{\infty }^{3}}{3k^{*}}\frac{\partial^{2}T}{\partial y^{2}}$$
the physical quantities used in Equations (1) to (9) are defined in the Nomenclature.

Now, the subsequent similarity transformations utilized can be defined as
(10)$$\left.\begin{array}{l} \eta =\sqrt{\frac{a}{\nu }}z,\;\;\;\;\;\;\;\;\;u=axf_{\eta }\left(\eta \right),\;\;\;\;\;\;\;\;\;v=ayg_{\eta }\left(\eta \right)\\ w=-\sqrt{a\nu }\left(f\left(\eta \right)+g\left(\eta \right)\right),\;\;\;\;\;\;\;\;\;\;\;\;\;\;\;\;\;\;\theta \left(\eta \right)=\frac{T-T_{\infty }}{T_{w}-T_{\infty }} \end{array}\right\}$$

We use these similarity transformations in Equations (1) to (4) to calculate the following ODEs
(11)$$\begin{array}{cc} \frac{\varepsilon_{1}}{\varepsilon_{2}}f_{\eta \eta \eta } & +f_{\eta \eta }\left(f+g\right)-f_{\eta }{}^{2}+\gamma f_{\eta }-\left(\frac{\varepsilon_{1}}{\varepsilon_{2}}Da^{-1}+\frac{\varepsilon_{3}}{\varepsilon_{2}}M\right)f_{\eta }\\ & +\beta \left\{f_{\eta \eta \eta \eta }\left(f+g\right)+f_{\eta \eta }\left(f_{\eta \eta }-g_{\eta \eta }\right)-2f_{\eta \eta \eta }\left(f_{\eta }+g_{\eta }\right)\right\}=0 \end{array}$$
(12)$$\begin{array}{cc} \frac{\varepsilon_{1}}{\varepsilon_{2}}g_{\eta \eta \eta } & +g_{\eta \eta }\left(f+g\right)-g_{\eta }{}^{2}+\gamma g_{\eta }-\left(\frac{\varepsilon_{1}}{\varepsilon_{2}}Da^{-1}+\frac{\varepsilon_{3}}{\varepsilon_{2}}M\right)g_{\eta }\\ & +\beta \left\{g_{\eta \eta \eta \eta }\left(f+g\right)+g_{\eta \eta }\left(g_{\eta \eta }-f_{\eta \eta }\right)-2g_{\eta \eta \eta }\left(f_{\eta }+g_{\eta }\right)\right\}=0 \end{array}$$
(13)$$\left(\varepsilon_{5}+R\right)\theta_{\eta \eta }+\mathrm{Pr}\varepsilon_{4}\theta_{\eta }\left(f+g\right)=0$$
Reduced boundary conditions are as follows,
(14)$$\left.\begin{array}{l} f_{\eta }\left(0\right)=d+Lf_{\eta \eta }\left(0\right),\;\;\;f\left(0\right)=V_{C},\;\;\;f_{\eta }\left(\infty \right)\to 0,\;\;\;f_{\eta \eta }\left(\infty \right)\to 0,\\ g\left(0\right)=c+Lg_{\eta \eta }\left(0\right),\;\;\;\;\;\;\;g_{\eta }\left(\infty \right)\to 0,\;\;\;\;\;\;\;\;\;\;\;\;\;\;\;g_{\eta \eta }\left(\infty \right)\to 0,\\ \theta_{\eta }\left(0\right)=-Bi\left(1-\theta \left(0\right)\right),\;\;\;\;\;\;\;\;\;\;\;\;\;\theta \left(\infty \right)\to 0, \end{array}\right\}$$
Here, $Da^{-1}=\frac{\mu_{f}}{\rho_{f}Ka},$ is an inverse Darcy number, $M=\frac{\sigma_{f}B_{0}{}^{2}}{\rho_{f}a},$ is a Hartman number, $\beta =\frac{ka}{\nu }$ is a viscoelastic parameter, $\gamma =\frac{\gamma_{0}}{a}$ indicates the parameter of porosity, $L=l\sqrt{\frac{a}{\nu }}$ is the first order velocity slip parameter, $V_{C}=-\frac{w_{0}}{\sqrt{a\nu }}$ indicates mass transpiration with $V_{C}>0$ denotes suction, $V_{C}<0$ indicates injection, $V_{C}=0$ for impermeable sheet, and $\mathrm{Pr}=\frac{{\left(\mu C_{P}\right)}_{f}}{\kappa_{f}}$ is a Prandtl number. $d\;\;\mathrm{and}\;\;c=\frac{b}{a}$ denotes the stretching/shrinking sheet parameters along the *x* and *y* axis, respectively. If d=1, it indicates stretching rate; if d=−1, it indicates shrinking rate. 

The nanofluid quantities used in Equations (11) to (13) can be defined as (see Afridi et al. [40] and Afridi and Qasim [41]):(15)$$\varepsilon_{1}=\frac{\mu_{nf}}{\mu_{f}},\;\;\;\varepsilon_{2}=\frac{\rho_{nf}}{\rho_{f}},\;\;\;\varepsilon_{3}=\frac{\sigma_{nf}}{\sigma_{f}},\;\;\;\varepsilon_{4}=\frac{{\left(\rho C_{P}\right)}_{nf}}{{\left(\rho C_{P}\right)}_{f}},\;\;\;\varepsilon_{5}=\frac{\kappa_{nf}}{\kappa_{f}}$$

Moreover, for our convenience, we use $\Gamma =\gamma -\left(\frac{\varepsilon_{1}}{\varepsilon_{2}}Da^{-1}+\frac{\varepsilon_{3}}{\varepsilon_{2}}M\right)$ in the further mathematical sequel, this term combines the magnetic interaction M>0, porosity parameter γ, and inverse Darcy number $Da^{-1}$.

## 2. Analytical Solutions

### 2.1. Analytical Solution of Momentum Equation

Based on the analytical solution derived in Crane [3], Aly [42], and Mahabaleshwar et al. [43], for some special cases of stretching sheet problems, we assume the solutions of Equations (11) and (12) are of the form
(16)$$f\left(\eta \right)=V_{C}+\frac{d\left(1-\exp\left(-\lambda \eta \right)\right)}{\lambda \left(1+L\lambda \right)}$$
(17)$$g\left(\eta \right)=\frac{d\left(1-\exp\left(-\lambda \eta \right)\right)}{\lambda \left(1+L\lambda \right)}$$

The solutions defined in Equations (16) and (17) satisfy all the boundary conditions defined in Equation (14), then substitute these solutions into Equation (11) at the limiting value η→∞ to find the following resulting equations:(18)$$\left(-\varepsilon_{1}+2\beta d\varepsilon_{2}\right)\lambda^{2}+2d\varepsilon_{2}=0$$
(19)$$-2d\left(1+\beta \lambda^{2}\right)\varepsilon_{2}+\left(1+L\lambda \right)\left(\gamma \varepsilon_{2}-\left(\varepsilon_{1}Da^{-1}+\varepsilon_{3}M\right)\right)-\lambda \left(V_{C}-\varepsilon_{1}\lambda +\varepsilon_{2}\beta V_{C}\lambda^{2}\right)=0$$

Solving above two solutions yields the following results:(20)$$\lambda =\sqrt{\frac{2\varepsilon_{2}}{2\beta \varepsilon_{2}-\varepsilon_{1}}}\;V_{C}=\frac{-2d\varepsilon_{2}\left(1+\beta \lambda^{2}\right)-\left(\varepsilon_{1}\Lambda +\varepsilon_{3}M-\gamma \varepsilon_{2}\right)\left(1+L\lambda \right)+\varepsilon_{1}\lambda^{2}\left(1+L\lambda \right)}{\lambda \left(1+\beta \lambda^{2}\right)\left(1+L\lambda \right)}$$

Further, the local skin friction coefficient can be determined as
(21)$$f_{\eta \eta }\left(0\right)=g_{\eta \eta }\left(0\right)=-\frac{\lambda d}{\left(1+L\lambda \right)}$$

### 2.2. Analytical Solution of Energy Equation 

Introducing a new variable ξ as
(22)$$\xi =\frac{\mathrm{Pr}}{\lambda^{2}}e^{-\lambda \eta },$$
on substituting Equation (22) in Equation (13) to achieve the equation
(23)$$\left(\varepsilon_{5}+R\right)\xi \frac{\partial^{2}\theta }{\partial \xi^{2}}+\left\{\left(\varepsilon_{5}+R\right)-\varepsilon_{4}\mathrm{Pr}\left(\frac{V_{C}\lambda \left(1+L\lambda \right)+2d}{\lambda^{2}\left(1+L\lambda \right)}\right)+\frac{2d\varepsilon_{4}}{1+L\lambda }\xi \right\}\frac{\partial \theta }{\partial \xi }=0$$
the boundary condition reduces to
(24)$$\frac{\mathrm{Pr}}{\lambda }\left(\theta_{\eta }\left(\frac{\mathrm{Pr}}{\lambda^{2}}\right)\right)=-Bi\left(1-\theta \left(\frac{\mathrm{Pr}}{\lambda^{2}}\right)\right),\;\theta \left(0\right)=0$$

After solving Equations $\left(23\right)$ and $\left(24\right)$, the solution of the energy equation becomes
(25)$$\theta \left(\eta \right)=\frac{Bi\;\Gamma \left(1-\frac{B}{C},0\right)-Bi\;\Gamma \left(1-\frac{B}{C},\frac{-2d\varepsilon_{4}\mathrm{Pr}}{\left(1+L\lambda \right)\lambda^{2}}\exp\left(-\lambda \eta \right)\right)}{\lambda {\left(\frac{-2d\varepsilon_{4}\mathrm{Pr}}{\left(1+L\lambda \right)\lambda^{2}}\right)}^{1-\frac{B}{C}}\exp\left(\frac{-2d\varepsilon_{4}\mathrm{Pr}}{\left(1+L\lambda \right)\lambda^{2}}\right)+Bi\left\{\Gamma \left(1-\frac{B}{C},0\right)-\Gamma \left(1-\frac{B}{C},\frac{-2d\varepsilon_{4}\mathrm{Pr}}{\left(1+L\lambda \right)\lambda^{2}}\right)\right\}}$$
where,
(26)$$\begin{array}{l} B=\left(\varepsilon_{5}+R\right)-\varepsilon_{4}\mathrm{Pr}a\\ C=\left(\varepsilon_{5}+R\right)\\ a=\frac{V_{C}\lambda \left(1+L\lambda \right)+2d}{\lambda^{2}\left(1+L\lambda \right)} \end{array}$$

## 3. Results and Discussions

Three-dimensional nanofluid flow due to a porous stretching/shrinking sheet with mass transpiration and radiation is examined in the current analysis. The resulting non-linear PDES are altered into ODEs with the help of similarity transformations, then the problem is verified analytically, and mass transpiration is solved under a special case. The energy equation is solved with a Biot number. Graphene nanofluid volume fraction is used to derive the problem analytically. The analytical solution of momentum and energy equation is indicated in Equations (20) and (25), respectively. In solution domain, λ can be linked through Equation (20), the mass transpiration depends on λ, Γ, β and L, and the temperature profile depends on $V_{C},\;\;R,\;\;\mathrm{Pr},\;\;\lambda \;\;\mathrm{and}\;d$. The physical scenario can be understood with the help of different physical parameters, then by using this we conclude the following discussion.

Figure 2a,b represents the effect of transverse velocity $f\left(\eta \right)$ verses similarity variable η for different choices of Γ for stretching and shrinking cases, respectively, with fixed the parameters as d=1, β=ϕ=0.1. Here, it is seen that the transverse velocity $f\left(\eta \right)$ decreases with an increase in the values of Γ. Here, the red solid lines portray the flow patterns at L=0 and black solid lines portray the flow patterns at L=0.5. From these figures, it seems that the boundary value thickness is wider for the shrinking sheet case compared to the stretching sheet case. Additionally, the velocity is higher for more values of L for the stretching sheet case, but this effect is reversed for the shrinking sheet case.

Figure 3a,b represents the impact of tangential velocity $f_{\eta }\left(\eta \right)$ on η for various values of β and ϕ respectively, keeping the parameters d=Γ=1, ϕ=0.1 at Figure 3a and d=Γ=1, β=0.1 at Figure 3b. From these figures, we can conclude that the tangential velocity $f_{\eta }\left(\eta \right)$ increases with increase in the values of β and ϕ. Here, the red solid lines also represent the flow patterns at L=0 and the black solid lines portray the flow patterns at L=0.5. Here, the unknown λ value linked with these parameters through Equation (20). Physically, the parameter Γ is the combination of a magnetic interaction, inverse Darcy number, and porosity parameter. Mathematically, it is represented as follows
(27)$$\Gamma =\gamma -\left(\frac{\varepsilon_{1}}{\varepsilon_{2}}Da^{-1}+\frac{\varepsilon_{3}}{\varepsilon_{2}}M\right)$$

This parameter controls the domain of existence and permits the presence of solutions for both wall transpirations for Γ<0, whereas mass suction corresponds to the certain values of Γ≥0. It is also observed that existence domain seems to be wider for the stretching sheet case as compared to the shrinking sheet case. Increasing slip or increasing viscoelasticity decreases the shear stress. In these figures, the impact of Γ helps to control the uniformity of the flow and to calculate the values of η. 

Equation (25) depicts the analytical expression of the energy equation along with a Biot number; this expression can provide the analytical solution of the temperature for Γ, R, ϕ & β; the thermal analysis is valid for Pr=6.2,  β<1, ϕ<1 and the slip parameter is fixed to L=1. According to the laminar boundary layer theory, the dependence of all these numbers and parameters are discussed below. Figure 4, Figure 5, Figure 6 and Figure 7 depict the impact of temperature profile $\theta \left(\eta \right)$ verses similarity variable η for different choices of different physical parameters. Figure 4a,b depict the impact of $\theta \left(\eta \right)$ verses η for various values of Γ at d=1  and  d=−1, respectively, with the fixed parameters ϕ=β=0.1, R=Bi=1. In both the stretching and shrinking cases, the $\theta \left(\eta \right)$ increases with an increase in the values of Γ. $\theta \left(\eta \right)$ increases with increases in the values of R for both d=1  and  d=−1 indicated, respectively, in Figure 5a,b. In this case, the other parameters are fixed to ϕ=Γ=β=0.1,  Bi=1. Figure 6a,b depict the impact of $\theta \left(\eta \right)$ verses η for various choices of ϕ; here, it is seen that $\theta \left(\eta \right)$ increases with increases of ϕ for both stretching and shrinking cases, and the other parameters are fixed to Γ=β=0.1, R=Bi=1. Figure 7a,b demonstrate the effect of temperature profile $\theta \left(\eta \right)$ verses η for various values of β, keeping the other parameters as ϕ=Γ=0.1, R=Bi=1. Here, it is seen that $\theta \left(\eta \right)$ increases with an increase in the values of β for both d=1  and  d=−1.

Figure 8a,b depict the impact of mass transpiration $V_{C}$ on solid volume fraction ϕ for various values of Γ. Domain $V_{C}$ moves towards negative values if we increase the values of Γ for both stretching and shrinking cases. Here, it seems to be that the shrinking case is wider than the stretching case.

The incomplete gamma function $\Gamma \left(a,z\right)$ becomes infinite when a=z=0. This knowledge is very significant for gaining the knowledge about Nusselt number and threshold parameters. This leads to the first non-zero heat transfer rate which is triggered by the incomplete gamma function; these results are not discussed much in our work (see Turkyilmazoglu [28]).

## 4. Concluding Remarks

An investigation has taken place on 3-D MHD graphene water nanofluid through porous media in the presence of mass transpiration and radiation. A closed form solution is obtained for both the flow and temperature, and mass transpiration is solved under a special case. Slip condition was also taken into account. The present work is useful in many real-life applications such automotive cooling systems, power generation, microelectronics, and air conditioning. 

By using this analysis, the following results can be concluded:Stretching case is wider than shrinking case.Velocity decreases with increases in the values of Γ.Tangential velocity $f_{\eta }\left(\eta \right)$ decreases with an increase in the values of β and ϕ.$\theta \left(\eta \right)$ is more for more values of Γ and R for both the stretching and shrinking cases.$\theta \left(\eta \right)$ is increases with increases in the values of ϕ and β in the stretching case and the shrinking case.The limiting parameters $Da^{-1}=\phi =R=0,\;\;\varepsilon_{1}\;\mathrm{to}\;\varepsilon_{5}=1,\;Bi\to \infty .$ in the present work is transformed into the work of Turkyilmazoglu [28] work.The classical Crane (1970) flow is recovered if the limiting parameters $M=\beta =Da^{-1}=\phi =R=L=\gamma =0,\;\;\varepsilon_{1}\;\mathrm{to}\;\varepsilon_{5}=1,\;\;\;Bi\to \infty .$



## Figures and Tables

**Figure 1 micromachines-13-00116-f001:**
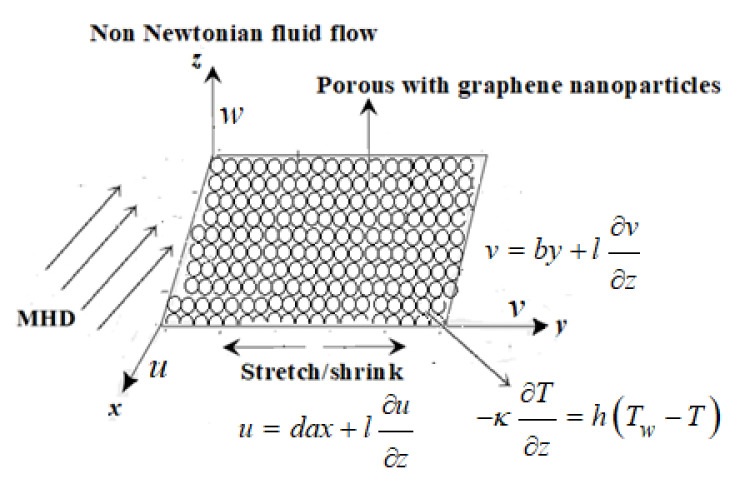
Physical diagram of three-dimensional fluid flow.

**Figure 2 micromachines-13-00116-f002:**
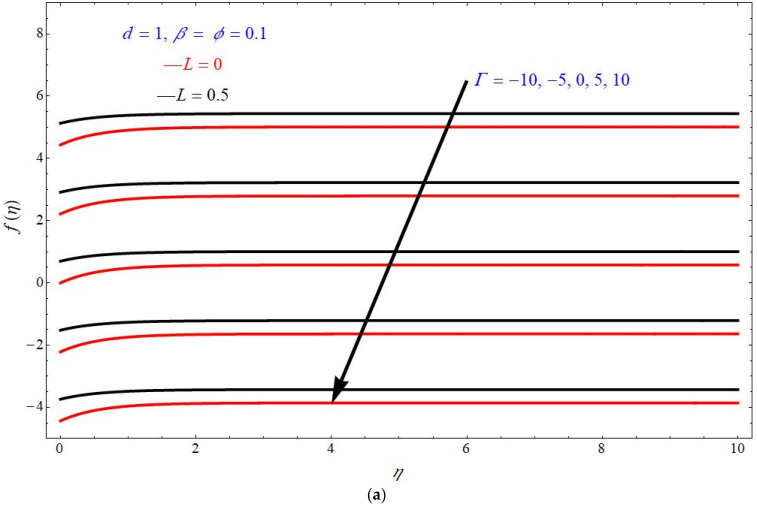
Effect of $f\left(\eta \right)$ on η for various choices of Γ (**a**) stretching and (**b**) shrinking cases.

**Figure 3 micromachines-13-00116-f003:**
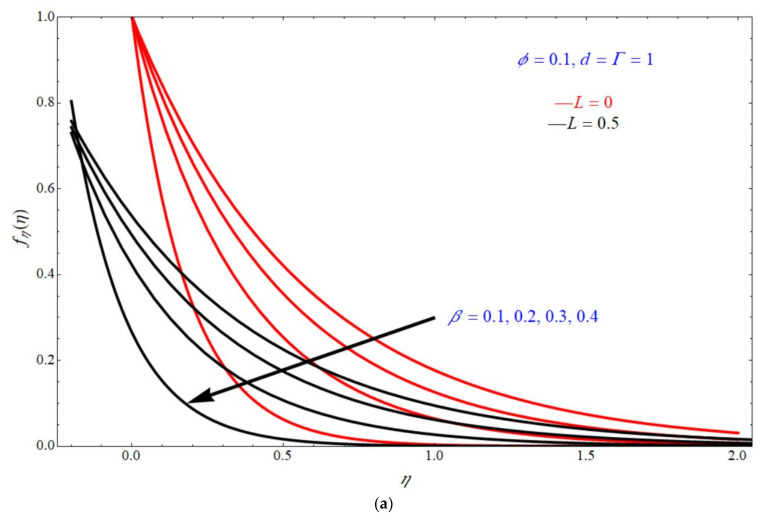
Impact of $f_{\eta }\left(\eta \right)$ on η for various choices of (**a**) β and (**b**) ϕ.

**Figure 4 micromachines-13-00116-f004:**
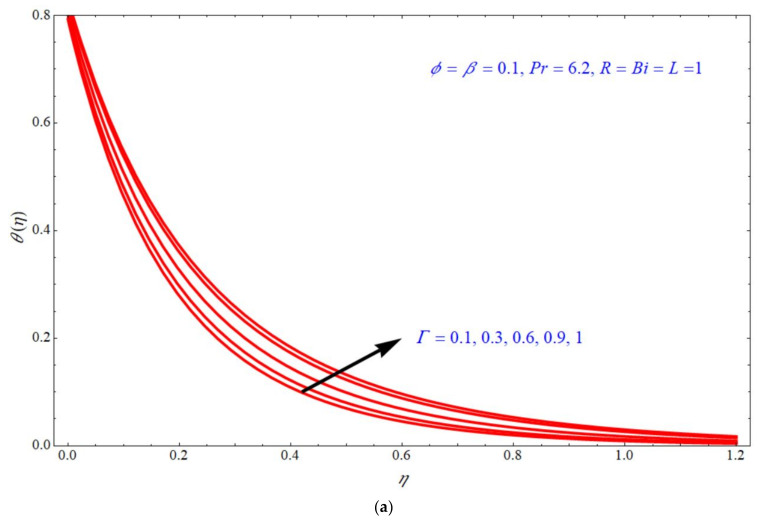
Impact of $\theta \left(\eta \right)$ on η for different choices of Γ at (**a**) d=1 and (**b**) d=−1.

**Figure 5 micromachines-13-00116-f005:**
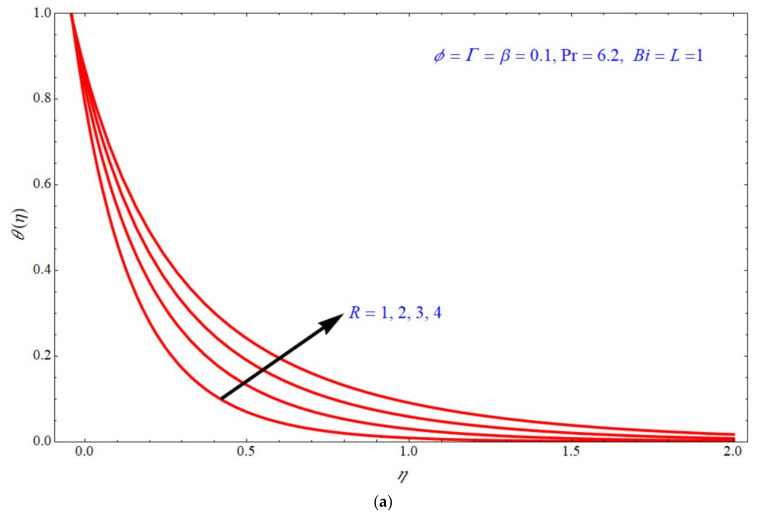
Impact of $\theta \left(\eta \right)$ on η for various choices of R at (**a**) d=1 and (**b**) d=−1.

**Figure 6 micromachines-13-00116-f006:**
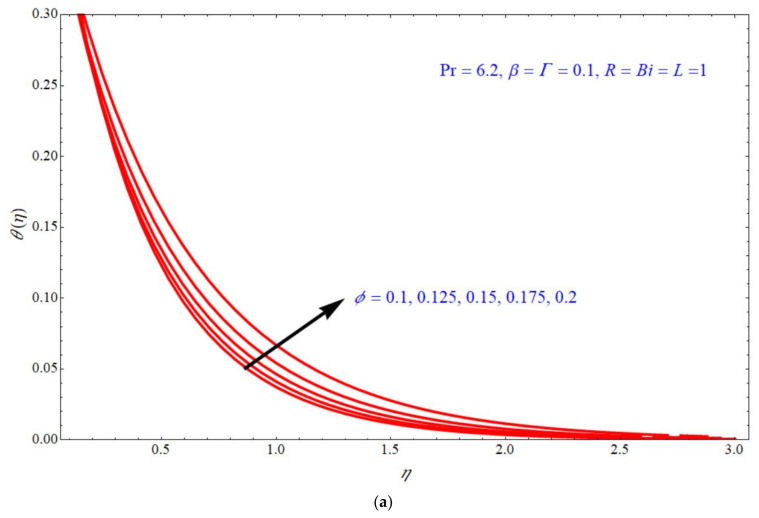
Impact of $\theta \left(\eta \right)$ on η for various choices of ϕ at (**a**) d=1 and (**b**) d=−1.

**Figure 7 micromachines-13-00116-f007:**
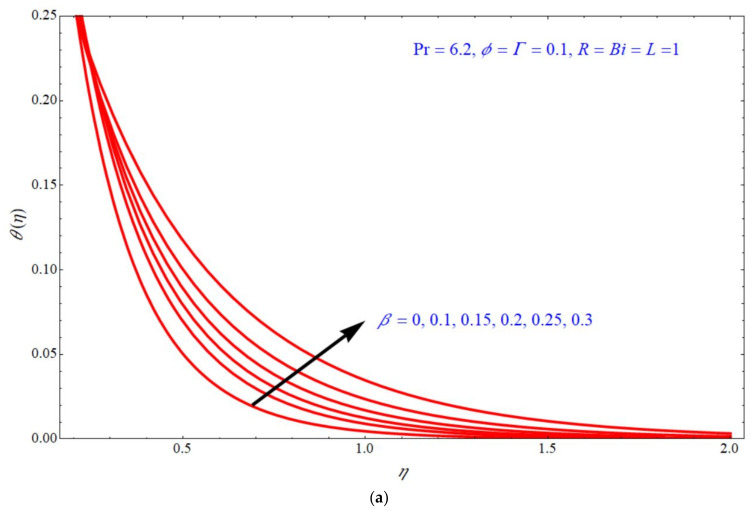
Impact of $\theta \left(\eta \right)$ on η for various choices of β at (**a**) d=1 and (**b**) d=−1.

**Figure 8 micromachines-13-00116-f008:**
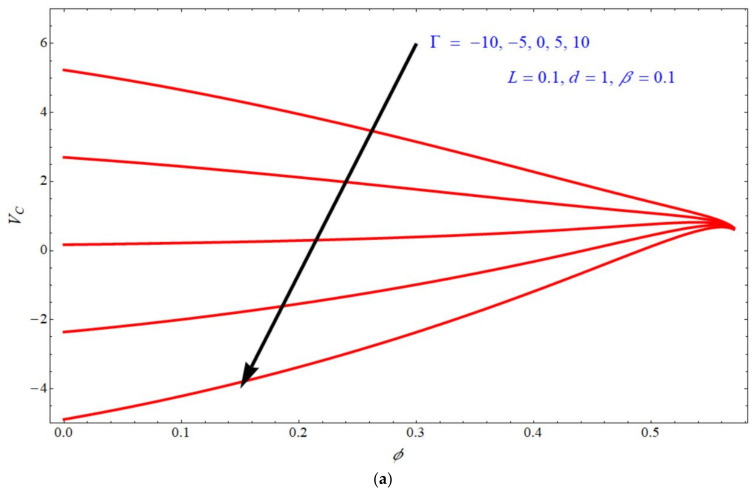
Impact of $V_{c}$ on ϕ for various choices of Γ at (**a**) d=1 and (**b**) d=−1.

**Table 1 micromachines-13-00116-t001:** Thermophysical properties of base fluid and nanoparticles.

	$C_{P}\;\left(J/kgK\right)$	$\rho \;\left(kg/m^{3}\right)$	$k\;\left(W/mK\right)$	$\sigma \;{\left(\Omega /m\right)}^{-1}$
Pure water $\left(H_{2}O\right)$	4179	997.1	0.613	0.05
Graphene (G)	2100	2250	2500	1 × 10^7^

## Data Availability

Data sharing is not applicable to this article.

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
