# Peer review of "An MHD Fluid Flow over a Porous Stretching/Shrinking Sheet with Slips and Mass Transpiration"

_micromachines, 2022, doi:10.3390/mi13010116_

Round 1

Reviewer 1 Report

Dear editor the subject of the manuscript lies in the scope of the journal but need substantial changes before acceptance. My detail comments are given below

  1. Rewrite the abstract in precise manner with two or three important findings included over there.
  2. Write down the novelty of the present study in the introduction.
  3. Specific real applications to the considered problem should be added in the conclusion section as well as in introduction.
  4. Comparison for limiting case of parameters need to be simulated.
  5. A nomenclature should be added with SI units.
  6. Improve the Quality of Figures. 
  7. Introduction part should be improved with latest relevant findings. Author (s) should use the following in this regard

          ::Analysis of three-dimensional stagnation point flow over a radiative surface.

         :: Second law analysis of three dimensional dissipative flow of hybrid nanofluid

         :: Entropy generation in three dimensional flow of dissipative fluid

         :: Entropy Generation in a Dissipative Nanofluid Flow under the Influence of Magnetic Dissipation and Transpiration

          ::Entropy generation analysis of spherical and non-spherical Ag-Water nanofluids in a porous medium with magnetic and porous dissipation.

          ::Comparative study and entropy generation analysis of Cu–H2O and Ag–H2O nanofluids flow over a slendering stretching surface

Author Response

Attached file 1

Reviewer 2 Report

Dear authors

Please incroporate the comments

Author Response

Attached file2

Reviewer 3 Report

The study is well reported and has some worth for publication in Micromachines. Some suggestions are listed below: 

  1. The title involves the word Transpiration. what does it mean? It's better to use mass transportation or mass flux instead of transpiration.
  2. The abstract is very short and it lacks the applications of the present study. 
  3. include some major results in the abstract. 
  4. The introduction is not sufficient. The authors must include some papers on MHD and also the Porous medium related literature that will strengthen the study. For example the papers published in recent years like 10.2174/1386207324666210903144447; 10.3390/mi12060605; 10.3390/mi12040374; 10.1007/s13204-020-01625-2; 10.1002/htj.22292; 10.1515/ijcre-2021-0109; 10.1115/1.4052985can be very handy to improve the literature. 
  5. The geometry diagram is not clear. Please include all the flow directions and boundary conditions in the diagram to make it more visible. 
  6. Equation 5 (u=dax + ... ) what does dax means? 
  7. Explain the constituent terms of radiative flux. 
  8. Add nomenclature table and list all the parameters involved in the study. 
  9. Discussion is not sufficient. Add more logical arguments besides the decreasing increasing behavior of the profiles. 
  10. Fig 2 (a,b) needs more explanation. 
  11. The authors talk about mass transportation but i can't see any data or graph for the drag force, heat or mass flux. Why? 

The paper is good and can be accepted for publication after above mentioned amendments. 

Author Response

Attached file3

Round 2

Reviewer 1 Report

Now, Paper can be accepted for publication

Reviewer 2 Report

This work is now ready to be published in Micromachine.

Reviewer 3 Report

Most of the comments are addressed. I accept it in present form.